# The Impact of Purple-Flesh Potato (*Solanum tuberosum* L.) cv. “Shadow Queen” on Minor Health Complaints in Healthy Adults: A Randomized, Double-Blind, Placebo-Controlled Study

**DOI:** 10.3390/nu14122446

**Published:** 2022-06-13

**Authors:** Mari Maeda-Yamamoto, Osamu Honmou, Masanori Sasaki, Akane Haseda, Hiroyo Kagami-Katsuyama, Toshihiko Shoji, Ai Namioka, Takahiro Namioka, Hirotoshi Magota, Shinichi Oka, Yuko Kataoka-Sasaki, Ryou Ukai, Mitsuhiro Takemura, Jun Nishihira

**Affiliations:** 1Food Research Institute, National Agriculture and Food Research Organization (NARO), Tsukuba 305-8642, Japan; tshoji@affrc.go.jp; 2Department of Neural Regenerative Medicine, Institute for Frontier Medical Sciences, School of Medicine, Sapporo Medical University, Sapporo 060-8556, Japan; honmou@sapmed.ac.jp (O.H.); msasaki@sapmed.ac.jp (M.S.); anamioka@tokyoh.johas.go.jp (A.N.); tnamioka@tokyoh.johas.go.jp (T.N.); hirotoshimagota@me.com (H.M.); soka@sapmed.ac.jp (S.O.); yuko.k.sasaki@sapmed.ac.jp (Y.K.-S.); ryou@sapmed.ac.jp (R.U.); light.extent@gmail.com (M.T.); 3Department of Medical Management and Informatics, Hokkaido Information University, Ebetsu 069-0832, Japan; a-haseda@s.do-johodai.ac.jp (A.H.); k.katsuyama@do-johodai.ac.jp (H.K.-K.); nishihira@do-johodai.ac.jp (J.N.)

**Keywords:** purple-fresh potato cv. “Shadow Queen”, anthocyanins, improvement of minor health complaints, reducing the effect of the stress response, the brief job stress questionnaire

## Abstract

The purple-flesh potato (*Solanum tuberosum* L.) cultivar “Shadow Queen” (SQ) naturally contains anthocyanins. This randomized, double-blind, placebo-controlled study determines whether ingesting purple potatoes increases the number of mesenchymal stem cells (MSC) and improves stress response, a minor health complaint in healthy adults (registration number: UMIN000038876). A total of 15 healthy subjects (ages: 50–70 years) with minor health complaints were randomly assigned to one of two groups. For 8 weeks, the placebo group received placebo potatoes cv. “Haruka” and the test group received test potato cv. SQ containing 45 mg anthocyanin. The MSC count and several stress responses were analyzed at weeks 0 and 8 of the intake periods. The ingestion of a SQ potato did not affect the MSC count but markedly improved psychological stress response, irritability, and depression as minor health complaints compared with “Haruka”. No adverse effects were noted. Hence, an 8-week intake of SQ could improve stress responses.

## 1. Introduction

In Japan, the proportion of people experiencing work-related stress and the nature of their stress are changing along with changes like the declining birthrate, aging population, mass retirement of baby boomers, the introduction of performance-based systems, intensifying international competition, escalating burdens caused by personnel cutbacks, and worsening economic conditions. Per the Ministry of Health, Labor and Welfare’s “Survey of Workers’ Health Status” conducted every 5 years, the fraction of workers who “feel stressed at work or in their professional life” was 50.6% (1982), 55% (1987), 57.3% (1992), 62.8% (1997), 61.5% (2002), 58% (2007), 60% (2008), 60.9% (2012), and 58.3% (2017), suggesting that approximately 60% of the working population is now stressed at work. The 2012 survey results by age group are 58.2% (20s), 65.2% (30s), 64.6% (40s), 59.1% (50s), and 46.9% (≥60s), with the so-called “prime working age” group in their 30s and 40s experiencing the highest levels of stress [1]; this trend is similar for both men and women.

Psychological stress reactions include low energy, irritability, anxiety, and depression (low mood, low interest). In contrast, physical stress reactions include joint pain, headache, stiff shoulders, back pain, eye fatigue, palpitations, shortness of breath, stomach pain, loss of appetite, constipation, diarrhea, insomnia, and many other symptoms. Furthermore, the decline in labor productivity and economic loss owing to presenteeism has been perceived as a problem. A study of four Japanese pharmaceutical companies estimated the average annual economic losses that are due to medical/pharmaceutical expenses per employee, absenteeism, and presenteeism were $1165, $520, and $3055, respectively, with the economic loss that is due to presenteeism being the highest [2]. Moreover, of the 34 diseases and conditions surveyed, neck pain, stiff shoulders, lack of sleep, back pain, dry eyes, and depression had the highest presenteeism costs [2]. Thus, to decrease the substantial economic losses that are due to presenteeism, it is essential to improve minor health complaints to prevent diseases and create a vibrant, healthy, and long-lived society.

The Japanese Consumer Affairs Agency (CAA) launched a new food labeling system for “Foods with Function Claims (FFC)” in April 2015. Under this system, companies and agricultural producers can independently evaluate and describe scientific evidence on health food benefits and functional properties to promote informed consumption. FFCs are foods that are registered with the secretary-general of the CAA as products whose labels bear health-related claims based on scientific evidence, as assessed by food business operators. Many FFCs (e.g., vegetables, mushrooms, grains, bananas, grapes, chocolate, beverages, and supplements) that are projected to contribute to alleviating stress with functional ingredients, such as gamma-aminobutyric acid (GABA), have been developed and are available in the market. A possible mechanism of action for the temporary relief of mental stress by oral GABA intake is its action on the autonomic nervous system from peripheral ganglia via GABA_A_ receptors. Moreover, anthocyanins and polyphenols found in many plants, such as bilberry, cassis, grape, and purple sweet potato, have various biological functions, such as antioxidative, anti-eyestrain, anti-cardiovascular, anti-inflammatory, and anticancer activities [3], improvement of the peripheral muscle blood flow [4], and improvement of mental function [5] and mood [6], although these functions have not been explored in human subjects.

Pigmented potatoes, such as purple-flesh potatoes, naturally contain various anthocyanins. Reportedly, purple potatoes contain additional polyphenolic compounds, such as chlorogenic acid and caffeic acid, than white potatoes [7]. In addition, purple potatoes have anti-inflammatory [8], intestinal epithelial differentiation-promoting [9], and antihepatotoxic [10] properties and have been reported to increase the activity of antioxidant enzymes, such as catalase, glutathione S-transferase, and glutathione reductase, in cholesterol-fed rats [11], and decreased serum lipid and cholesterol levels [12]. Furthermore, purple potatoes improved the activity of antioxidant enzymes in leukocytes and reduced the malondialdehyde content in plasma in streptozotocin-induced diabetic rats [13].

“Shadow Queen” is a variety of purple-flesh potato high in anthocyanins [14], which is used to make snacks because of its bright purple color. A human short-term intervention study suggested a potential effect of anthocyanin-rich potato consumption on arterial stiffness [15]. In contrast, our previous study in rodents demonstrated that long-term treatment with “Shadow Queen” might increase the number of blood mesenchymal stem cells (MSC; unpublished).

Hence, the research question was to elucidate whether continuous intake of anthocyanin-containing purple potatoes improves stress responses or peripheral blood MSC count compared with continuous intake of anthocyanin-free potatoes in healthy Japanese adults (aged 50–70 years) experiencing stress.

## 2. Materials and Methods

### 2.1. Study Design

This was a double-blind, randomized, placebo-controlled, parallel-group study. Subjects were randomly assigned to either a test or placebo potato group with an allocation ratio of 1:1. Figure 1 shows the study protocol. The subjects were recruited from the Hokkaido Information University in Hokkaido (Japan) and were fully informed regarding the content and methods of this study. A week prior to the start of the test-food consumption was the washout period, and the subjects consumed the test food for 8 weeks. Each subject was tested on the screening day, on the first day of test food intake, 4 weeks after intake, and 8 weeks after intake. Screening for the first volunteers started on 19 November 2019, and the study was completed in March 2020.

### 2.2. Subjects

We enrolled 40 healthy volunteers (age: 50–69 years), of which 15 subjects (4 men, 11 women) were eligible. Appendix A lists the key eligibility and exclusion criteria. The 15 eligible subjects were randomly assigned to the “Shadow Queen” as the test group or “Haruka” as the placebo group stratified by gender, age, and visual analog scale (VAS) questionnaires. All assignments were computer generated using stratified block randomization at a third-party data center. Of note, doctors, nurses, clinical research coordinators, and statistical analyzers were blinded to the assignment information during this trial period; this information was disclosed only after the laboratory and analytical data were fixed, and the statistical analysis method was finalized.

### 2.3. Test Samples

The potatoes were grown in Hokkaido (Japan). Potatoes cv. “Shadow Queen” and “Haruka” were used as the test and placebo groups, respectively, in this study. Subjects ate potatoes (about 75 g) cooked in a microwave oven for 2 min once a day. Only potatoes were cooked. The timing of eating was determined separately for each subject. Seasonings (such as salt, butter, mayonnaise, and dressing) were not restricted, but the subjects were asked to record the name and amount of seasoning used in their daily log. The potatoes were analyzed for nutrients and anthocyanins at the Japan Food Research Laboratories (Tokyo, Japan). 

Anthocyanin content was measured by the colorimetric method, water by the atmospheric heating drying method, protein by the combustion method, fat by the acid hydrolysis method, ash by the direct ashing method, dietary fiber by the enzymatic-gravimetric method, and sodium by atomic absorption spectrophotometry. The conversion factor of nitrogen−protein was set at 6.25. Carbohydrates were calculated as “100 − (protein + fat + ash)”, sodium chloride (NaCl) as “sodium × 2.54”, and energy using energy conversion factors (protein: 4, fat: 9, carbohydrates: 4, and dietary fiber: 2).

Table 1 summarizes the analytical results. Of note, remarkably low analytical values are shown in Table 1 because these are expressed in delphinidin equivalents. The anthocyanins in potatoes differed from those contained in colored sweet potatoes, tea leaves, berries, and black rice, in that they are aglycones of petunidin and peonidin. “Shadow Queen” contains 72% petunidin 3-p-coumaroylrutinoside-5-glucoside (petanin), 14% peonidine-3-p-coumaroylrutinoside-5-glucoside (peonanin), 11% others, and 3% peonidine-3-caffeoyl rutinoside-5-glucoside [14]. The total anthocyanin content of “Shadow Queen”, calculated by comparison with the standard petanin, is 816 mg/100 g of fresh tuber [14].

### 2.4. Study Outcomes

The primary outcomes were defined as the peripheral blood MSC count. The secondary outcomes were defined as stress responses, answers to “The Brief Job-Stress Questionnaire (BJSQ)” for liveliness, irritability, fatigue, depression, physical complaints, psychological stress response, and physical stress response as minor health complaints, VAS questionnaire for fatigue, urinary 8-hydroxydeoxyguanosine (8-OHdG) levels, and blood 1,5-anhydroglucitol (1,5-AG).

### 2.5. Measurement of Peripheral Blood MSC Count

From each subject, 10 mL of peripheral blood was collected, and red blood cells (RBC) were lysed using 60 mL of RBC lysis solution (#158904; Qiagen, Hilden, Germany) for 10 min at room temperature. Then, the samples were centrifuged for 5 min (2300× *g*) at 20 °C, two times and diluted to 16 mL Dulbecco’s modified Eagle’s medium (DMEM; Sigma, St. Louis, MO, USA) supplemented with 30% heat-inactivated fetal bovine serum (FBS; Thermo Fisher Scientific Inc., Waltham, MA, USA), 2 mM L-glutamine (Sigma, St. Louis, MO, USA), 100 U/mL penicillin, 0.1 mg/mL streptomycin (Thermo Fisher Scientific Inc., Waltham, MA, USA), and cultured on 10 cm^2^ dish (5%, carbon dioxide (CO_2_), 37 °C) for 11 days. Then, all the adherent cells were detached with trypsin–EDTA (ethylenediaminetetraacetic acid) solution (Sigma, St. Louis, MO, USA) and manually counted using a hemocytometer (C-Chip; NanoEntek, Seoul, Korea).

### 2.6. Measurement of the BJSQ

In this study, stress responses were measured using the BJSQ recommended by the Ministry of Health, Labor and Welfare in Japan to check one’s stress status under the Industrial Safety and Health Act. Based on the 29 questions in the BJSQ, scores for liveliness, irritability, fatigue, anxiety, depression, physical complaints, psychological stress response, and physical stress response were calculated, and the degree of stress was judged according to these scores. In our paper (in submitting), we have demonstrated that the BJSQ can be used to assess minor health complaints and that the use of the items for liveliness, irritability, fatigue, and body complaints is mainly effective in assessing minor health complaints.

### 2.7. Measurement of VAS Questionnaire for Fatigue

We assessed the changes in stress responses before and after (0, 4, and 8 weeks) the intake using a VAS test for “eyestrain”, “eye pain”, “irritability”, and “stress”. The test was labeled “the worst condition” (0) on the left and “the best condition” (100) on the right on the extreme ends of a 100 mm line, and the subjects were asked to evaluate their condition when the questionnaire was administered by writing an “×” on the line. The questionnaire results were scaled by measuring the length from the left end to the “×” mark, and the values at each evaluation point were used. The questionnaire results were scaled by measuring the length from the left edge to the “×” mark, which was the value at each evaluation point.

### 2.8. Measurement of Hematological, Biological, and Diet Survey

Blood samples were obtained for testing at the baseline and weeks 4 and 8 of the study periods. Besides a medical interview by a doctor, each subject’s body composition (body weight (BW), body mass index (BMI), and body fat ratio (BFR)), and blood pressure (BP) were measured. General blood tests comprised lipid profile (triglyceride, total cholesterol, high-density lipoprotein cholesterol (HDL-C), and low-density lipoprotein cholesterol (LDL-C)); blood glucose profile (fasting blood glucose (FBG) and hemoglobin A1c (HbA1c)); complete blood count, including white blood cells (WBC), RBC, hemoglobin (Hb), hematocrit (Ht), and platelet count (Plt); liver function, including aspartate aminotransferase (AST), alanine aminotransferase (ALT), gamma-glutamyl transpeptidase (GTP), alkaline phosphatase (ALP), and lactate dehydrogenase (LDH); and renal function, including blood urea nitrogen, creatinine (CRE), and uric acid (UA). Moreover, Sapporo Clinical Laboratory, Inc. (Sapporo, Japan) conducted hematological and biological examinations. Each subject’s body composition was measured with a body composition analyzer DC-430A (Tanita Corp., Tokyo, Japan). Furthermore, urinary o-hydroxy-2′-deoxyguanosine (8-OHdG) was measured by enzyme-linked immunosorbent assay (ELISA) as one of the stress markers, and dietary intake and nutrients were assessed using the food frequency questionnaire (FFQ).

### 2.9. Ethics Committee

This study protocol was approved by the Ethics Committee of the Hokkaido Information University (Ebetsu, Hokkaido, Japan) and Sapporo Medical University (Sapporo, Hokkaido, Japan) per the principles of the Declaration of Helsinki (approval date October 30, 2019; approval number: 2019-29; approval date 9 December 2019; approval number: 1-2-57, respectively). This study was registered at www.umin.ac.jp/ctr/index.htm (registration number: UMIN000038876; date of registration: 13 December 2019). Written informed consents were obtained from all subjects before enrollment.

### 2.10. Statistical Analysis

Statistical analysis was performed using the per-protocol analysis set. A histogram plot was used for normality checking. In addition, differences between intervention groups were evaluated using Student’s *t*-test. For each group, differences between pre- and post-intake were evaluated using paired *t*-tests. In this study, all statistical analyses were performed using SPSS ver. 25 (IBM Japan, Tokyo, Japan). A *p* value of <0.05 was considered significant. Multiplicity was not controlled. Missing values were not complemented. The sample size was not calculated because no similar studies were available for reference.

## 3. Results and Discussion

### 3.1. Number of Subjects and Intake Rate of Test Potatoes

A total of seven subjects in the test group and eight in the placebo group completed the final visit assessment, and the data of these subjects were used for the final analysis. The average intake rates for the experimental period were 100% ± 0.0% and 98.7% ± 2.7%, as calculated using the intake check sheets of the test group and placebo group, respectively, as shown in Table 2. Figure 2 presents the flow diagram for this study. The food survey using the FFQ revealed no change in diet during the test period in either group (Appendix A).

### 3.2. Physical Characteristics

Table 2 presents the mean age, height, BW, BMI, BFR, VAS questionnaire, and intake rate for each group; these data did not markedly differ between the two groups, validating the appropriate allocation of subjects to the two groups.

### 3.3. Primary Outcomes

First, we evaluated the effect of enriched anthocyanin-containing potato “Shadow Queen” on the MSC count. Figure 3 shows a box-and-whisker plot of the changes in the MSC count from the baseline to week 8. The mean ± standard deviation (SD) of “Shadow Queen” and “Haruka” were 50,000 ± 280,595 and −44,286 ± 727,412, respectively. No significant difference was observed in the change in the MSC count between the test and placebo groups.

### 3.4. Secondary Outcomes

#### 3.4.1. Effect of “Shadow Queen” on Stress Responses Measured Using the BJSQ

Table 3 and Figure 4 show the changes in stress scores using the BJSQ as a secondary outcome after ingesting “Shadow Queen” and “Haruka” potatoes. The psychological stress responses (Figure 4A), irritability (Figure 4B), and depression (Figure 4C) scores were markedly lower in the test group than that of the placebo group after 8 weeks of intake. These results suggested that the intake of “Shadow Queen” might relieve stress.

Of note, GABA reportedly lowers BP and alleviates transient mental stress, activates GABA receptors in the peripheral ganglia, and inhibits the release of vasoconstrictor noradrenaline, thereby suppressing the sympathetic nervous system and lowering BP [15,16]. Moreover, it alleviates mental stress by inhibiting the sympathetic nervous system and increasing the parasympathetic nervous system [17,18,19]. The health benefits of anthocyanins are established, especially in preventing diseases related to oxidative stress, such as cardiovascular and cerebrovascular diseases [20]. A crucial function of flavonoids, including anthocyanins, is to lower cardiovascular, or BP, and enhance endothelial function. A study of the continuous consumption of cooked purple potatoes containing 288 mg of anthocyanins or white potatoes containing little or no anthocyanins in healthy adults reported a significant decline in the pulse wave velocity after 14 days of consuming purple potatoes (*p* = 0.001) [21]. In addition, daily intake of anthocyanin-rich boysenberry juice was beneficial for decreasing systolic BP in subjects with higher systolic BP [22]. Moreover, high-flavanol cocoa increased the forearm blood flow at rest and during stress, and the stress-induced cardiovascular and BP responses were similar in both conditions [23]. Flavanols are assumed to counteract endothelial dysfunction caused by mental stress and improve the peripheral blood flow during stress [23,24]. Reportedly, 72% of “Shadow Queen” anthocyanins have petunidin as an anthocyanidin. Notably, anthocyanidins, similar in structure to flavanols, are also expected to exhibit similar effects.

A systematic review of clinical trials considers the impact of food anthocyanin intake (55.8–877 mg/day) on acute and long-term cognitive function to be promising [25]. The longitudinal cross-sectional and co-twin analyses showed that higher anthocyanin intake was associated with improvements in age-related cognitive scores over a 10-year period. Specifically, higher anthocyanin intake was significantly associated with improved executive function and faster and simpler reaction times [26]. Electroencephalography (EEG) data of the acute administration of anthocyanin-rich blackcurrant juice (500 mg of polyphenols) revealed an anxiolytic effect, as indexed by the suppression of the α spectral power and an increase in the slow wave δ and θ spectral powers [27]. Since “Shadow Queen” contains anthocyanins, it is expected to have anti-anxiety and cognitive function benefits.

#### 3.4.2. The Effect of “Shadow Queen” on the VAS Questionnaire

We compared the changes listed in the VAS questionnaire for fatigue—“eyestrain”, “eye pain”, “irritability”, and “stress”—from the baseline for each week in both groups. No significant difference was observed between the test and placebo groups, as shown in Table 4. Nevertheless, the change in eye pain, irritability, and stress measurements after 8 weeks exhibited an increase in the test group (improvement in condition) and a decline in the placebo group (worsening of condition). Specifically, the stress score tended to have a higher score in the test group than in the placebo group (*p* = 0.058).

#### 3.4.3. The Effect of “Shadow Queen” on Another Secondary Endpoint

Table 4 presents the changes in 1,5-AG and 8-OHdG as a secondary outcome after taking “Shadow Queen” and “Haruka”. Among them, at week 8, the change in 1,5-AG levels in the test group was significantly lower than in the placebo group (*p* = 0.017); however, urinary 8-OHdG, a marker of oxidative stress, did not differ between the groups, as shown in Table 5.

Reportedly, a randomized study examined the effects of pigmented potato consumption on oxidative stress and inflammation biomarkers [28] in adult males and free-living healthy men (age: 18–40 years; *n* = 12/group) who consumed 150 g of cooked white potatoes (anthocyanin content; 0.0 mg/day), yellow potatoes (anthocyanin content; ~9 mg/day), or purple-flesh potatoes (anthocyanin content; ~186 mg/day) once per day for 6 weeks; the 8-OHdG concentration was lower in men who consumed either yellow potatoes or purple-flesh potatoes compared with white potatoes (*p* < 0.05) [29]. However, this effect was not due to anthocyanins, as it was also observed in yellow potatoes with low anthocyanin content. Our study suggests that the lack of reduction in oxidative stress markers in the test group could be owing to the lower anthocyanin content compared with purple-flesh potatoes.

### 3.5. Safety Assesment

We assessed the complete body composition, BP, blood cell counts, lipid parameters, liver function, and renal function after potato ingestion. As shown in Appendix A, the changes in the BFR and LDL-C from 0 weeks to 8 weeks after intake was significantly higher in the test group than in the placebo group (*p* = 0.039 and 0.028, respectively). In addition, the change in LDH from 0 weeks to 8 weeks after intake was significantly lower in the test group than in the placebo group (*p* = 0.049). The change in serum uric acid from 0 weeks to 4 weeks after intake was significantly higher in the test group than in the placebo group (*p* = 0.002). Nevertheless, each of the values usually remained within the normal range for the Japanese population (LDL-C: <140 mg/dL; LDH: 120–240 U/L; uric acid: <7.0).

We observed no serious adverse events in this study. One subject in the placebo group exhibited an adverse event related to diarrhea. In the test group, nine adverse events were reported, including elevated LDL-C (*n* = 1), tooth decay (*n* = 1), toothache-related symptoms (*n* = 1), tinnitus (*n* = 1), back pain and cramps (*n* = 1), sensation of the appearance of chronic alopecia areata (*n* = 1), linear urticaria of the upper body (*n* = 1), weakening of the pulse and nausea (*n* = 1), and orbital pain (*n* = 1). Accordingly, the principal investigator judged that no serious adverse event was related to the study intervention. Overall, these results suggested that the intake of “Shadow Queen” potato containing 45 mg of anthocyanins for 8 weeks is safe.

This study had a limitation that should be noted and, thus, findings of this study should be interpreted with much caution. The small sample size of the intervention limits the generalizability of the results and may have limited our ability to detect statistical significance. The present study also had a strength. The use of different questionnaires (the brief job-stress questionnaire and VAS) on stress responses to detect minor health complaints allowed us to determine which questionnaire provided better results. We think that this study provides important insights for similar intervention trials in the future.

## 4. Conclusions

In this study, the consecutive ingestion of the potato cultivar “Shadow Queen” for 8 weeks markedly improved psychological stress response, irritability, and depression in the BJSQ compared with “Haruka”. The stress score on the VAS tended to improve in the test group compared to the placebo group. Overall, this study indicates the potential “Shadow Queen” as a stress-improving agent. However, the findings should be interpreted very cautiously as the present study has a number of limitations. In the future, further studies with larger sample sizes are warranted to validate the stress-improving effects of more varieties of anthocyanin-rich potatoes.

## Figures and Tables

**Figure 1 nutrients-14-02446-f001:**
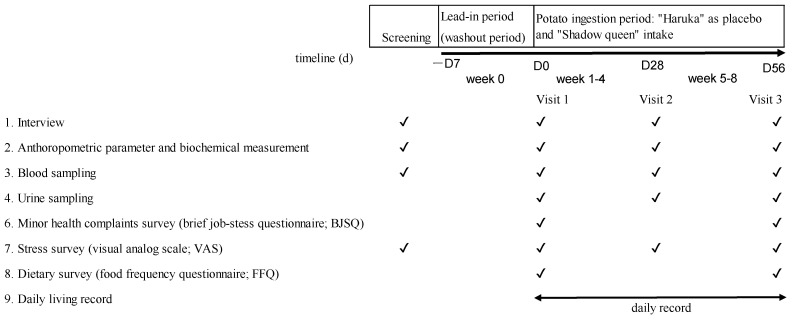
Schematic representation of the study/protocol.

**Figure 2 nutrients-14-02446-f002:**
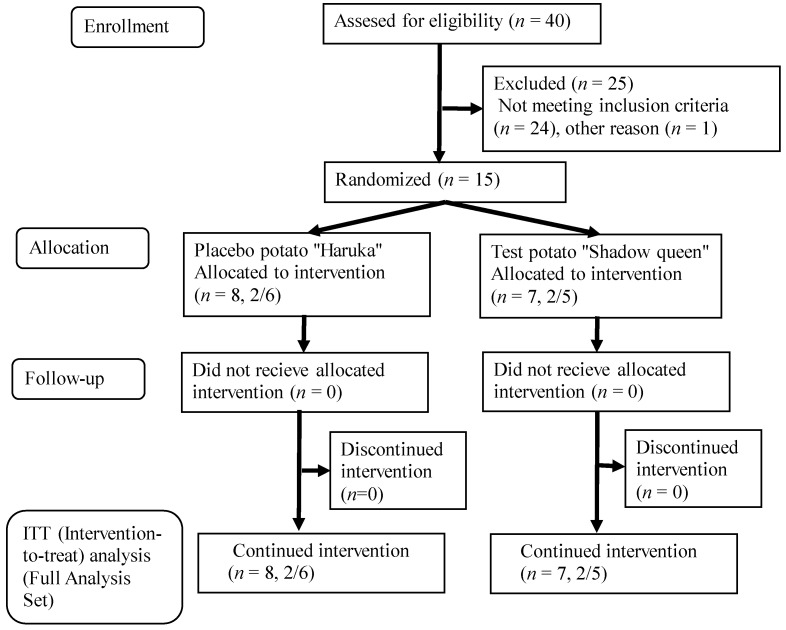
Study flowchart (*n* = total participants, male/female).

**Figure 3 nutrients-14-02446-f003:**
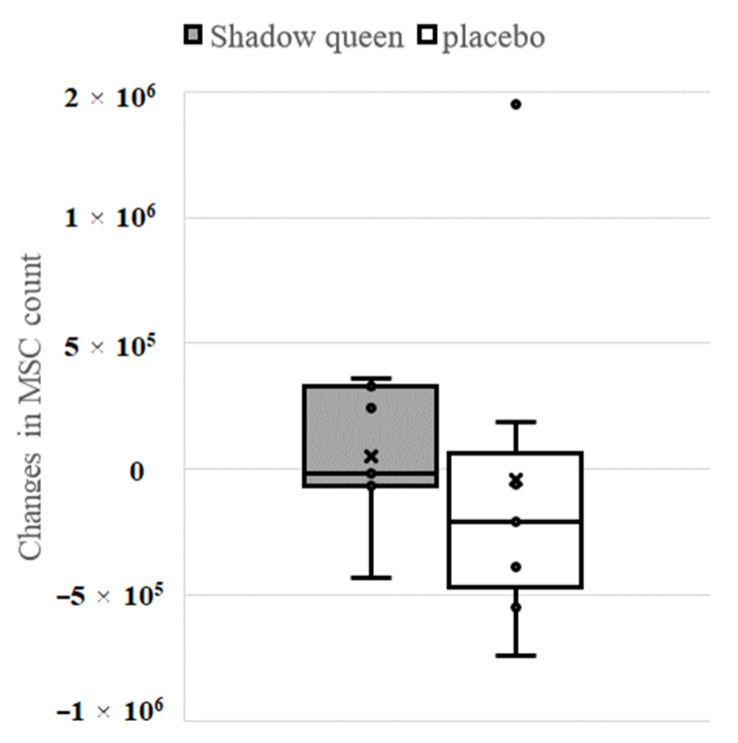
Change in the mesenchymal stem cells (MSC) count. The horizontal line in the boxplot represents the median, and the × mark represents the mean.

**Figure 4 nutrients-14-02446-f004:**
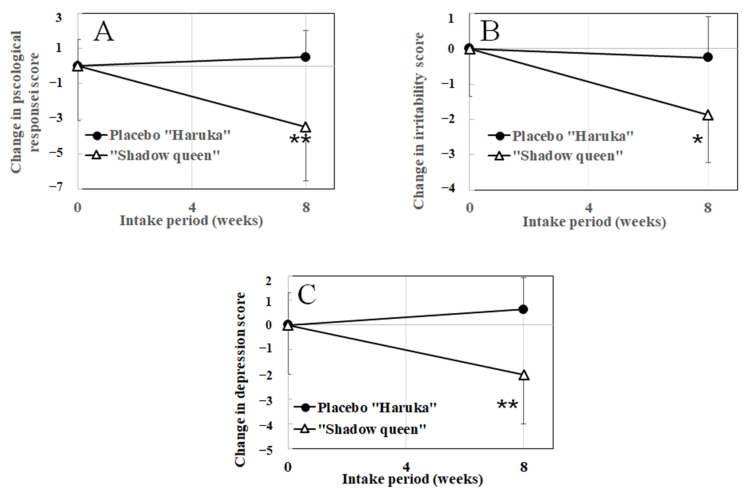
The changes in the brief job-stress questionnaire from week 0 to week 8. (**A**) Changes in the psychological stress response; (**B**) changes in irritability; and (**C**) changes in depression. Statistical significance: * *p* < 0.05; ** *p* < 0.01 vs. placebo group.

**Table 1 nutrients-14-02446-t001:** Nutritional and functional components in test potatoes (75 g/day).

	Placebo (“Haruka”)	Test (“Shadow Queen”)
Anthocyanins as delphinidin equivalents (mg)	0.0	45.0
Water (g)	60.2	58.4
Protein (g)	1.5	1.6
Fat (g)	0.15	0.15
Ash (g)	0.7	0.8
Carbohydrate (g)	12.5	14.1
Dietary fiber (g)	1.3	1.2
Energy (kcal)	55 kcal	62 kcal
NaCl (mg)	0.0	2.1

NaCl, sodium chloride.

**Table 2 nutrients-14-02446-t002:** Baseline characteristics of subjects in the test (“Shadow Queen”) and placebo (“Haruka”) groups.

Baseline Characteristics	Placebo (“Haruka“)	“Shadow Queen”	*p* Value
Gender (male/female) (*n*)	8 (2/6)	7 (2/5)	
Age (years)	56.1 ± 4.4	55.1 ± 3.8	0.653
Body weight (kg)	54.9 ± 10.3	50.9 ± 6.3	0.390
Body mass index (kg/m^2^)	22.6 ± 5.6	20.8 ± 2.2	0.199
Body fat ratio (%)	29.1 ± 5.6	25.3 ± 9.3	0.355
VAS questionnaires (mm)	196.1 ± 93.4	222.0 ± 97.6	0.610
Intake rate (%)	100.0 ± 0	98.7 ± 2.7	0.379

Values are shown as mean ± standard deviations. *n*, number of subjects; VAS (visual analog scale) questionnaires = mean of the sum of four VAS questionnaires (eye fatigue, eye pain, irritability, and awareness of stress). *p* values were determined by Student’s *t*-test for independent samples vs. placebo “Haruka”.

**Table 3 nutrients-14-02446-t003:** Changes in stress response scores as a secondary outcome after taking “Shadow Queen” and “Haruka”.

Stress Response (Score)	Intervention	Week 0	Week 8	Week 0 vs. Week 8	Changes in Placebo vs. Test Food
				*p* Value	*p* Value
Psychological stress response	“Haruka”	4.35 ± 3.77	4.75 ± 4.20	0.381	0.007 **
(0–18)	“Shadow Queen”	6.57 ± 4.35	3.14 ± 3.34	0.026^#^	
Physical stress response (0–11)	“Haruka”	2.50 ± 2.51	1.88 ± 2.59	0.370	0.794
	“Shadow Queen”	1.71 ± 2.43	0.86 ± 1.21	0.172	
Vigor (3–12)	“Haruka”	5.88 ± 2.59	6.00 ± 2.14	0.802	0.367
	“Shadow Queen”	6.00 ± 2.16	6.86 ± 3.02	0.225	
Irritability (3–12)	“Haruka”	6.25 ± 1.04	6.00 ± 1.31	0.563	0.028 *
	“Shadow Queen”	7.00 ± 2.77	5.14 ± 2.34	0.011 ^#^	
Fatigue (3–12)	“Haruka”	6.25 ± 2.60	5.13 ± 3.04	0.065	0.583
	“Shadow Queen”	7.14 ± 2.67	5.57 ± 2.82	0.042 ^#^	
Anxiety (3–12)	“Haruka”	5.13 ± 1.46	5.50 ± 2.62	0.634	0.146
	“Shadow Queen”	6.00 ± 1.63	4.86 ± 1.07	0.103	
Depression (6–24)	“Haruka”	9.38 ± 4.03	10.00 ± 4.24	0.217	0.009 **
	“Shadow Queen”	10.86 ± 2.73	8.86 ± 1.57	0.038 ^#^	
Physical Complaints (11–44)	“Haruka”	19.63 ± 6.21	18.38 ± 7.67	0.454	0.519
	“Shadow Queen”	19.43 ± 6.02	16.86 ± 3.80	0.063	

The brief job-stress questionnaire was used to examine stress responses. Values are shown as the means ± standard deviation. Student’s *t*-test for independent samples vs. placebo, paired *t*-test vs. 0 weeks was performed to analyze the values. Statistical significance, *; <0.05, **; <0.01 vs. placebo, #; <0.05 vs. 0 weeks.

**Table 4 nutrients-14-02446-t004:** Changes in the visual analog scale (VAS) on fatigue as a secondary outcome after taking “Haruka” or “Shadow Queen”.

Visual Analog Scale	Intervention	Week 0	Week 4	Week 8	ΔWeek 4	ΔWeek 8
Eyestrain (0–100)	“Haruka”	35.38 ± 23.87	42.75 ± 30.58	45.50 ± 23.78	7.38 ± 24.70	10.13 ± 21.52
	“Shadow Queen”	51.29 ± 32.63	67.86 ± 25.06	74.86 ± 22.66	16.57 ± 27.55	23.57 ± 29.22
	*p* value	-	-	-	0.507	0.324
Eye pain (0–100)	“Haruka”	70.00 ± 30.74	68.75 ± 32.67	67.38 ± 29.90	−1.25 ± 32.06	−2.63 ± 21.32
	“Shadow Queen”	76.14 ± 31.03	79.86 ± 26.12	86.14 ± 22.57	3.71 ± 29.64	10.00 ± 24.67
	*p* value	-	-	-	0.762	0.307
Irritability (0–100)	“Haruka”	66.38 ± 25.14	52.00 ± 20.28	58.50 ± 23.31	−14.38 ± 26.10	−7.88 ± 23.39
	“Shadow Queen”	65.43 ± 26.74	74.86 ± 28.35	73.00 ± 25.94	9.43 ± 27.16	7.57 ± 21.75
	*p* value	-	-	-	0.107	0.210
Stress (0–100)	“Haruka”	48.00 ± 31.06	43.00 ± 28.06	39.50 ± 21.21	−5.00 ± 23.10	−8.50 ± 17.31
	“Shadow Queen”	56.43 ± 30.99	72.29 ± 29.21	69.29 ± 22.76	69.29 ± 22.76	12.86 ± 22.39
	*p* value	-	-	-	0.158	0.058

A VAS questionnaire on fatigue was used, with self-assessment of the worst (0) to best (100) status. Values are shown as the means ± standard deviation. Student’s *t*-test for independent samples vs. placebo, paired *t*-test vs. 0 weeks was performed to analyze the values. Δ, changes.

**Table 5 nutrients-14-02446-t005:** Changes in the secondary outcome after taking “Shadow Queen” and “Haruka”.

Secondary Outcome	Intervention	Week 0	Week 8	Week 0 vs. Week 8	Changes in Placebo vs. Test Food
				*p* Value	*p* Value
Blood 1,5-anhydroglucitol(1,5-AG)	“Haruka”	18.30 ± 7.93	19.61 ± 8.84	0.012 ^#^	0.017 *
(µg/mL)	“Shadow Queen”	20.96 ± 3.97	20.54 ± 4.46	0.447	
Urine 8-hydroxy-2’-deoxyguanisine	“Haruka”	6.39 ± 1.59	5.28 ± 2.02	0.059	0.520
(8-OHdG) (ng/mg/Creatinine correction value)	“Shadow Queen”	7.33 ± 3.49	5.16 ± 2.93	0.194	

Values are shown as the means ± standard deviation. Student’s *t*-test for independent samples vs. placebo, paired *t*-test vs. 0 weeks was performed to analyze the values. Statistical significance, *; <0.05 vs. placebo, #; <0.05 vs. 0 weeks.

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
