# Peer review of "The Impact of Purple-Flesh Potato (Solanum tuberosum L.) cv. “Shadow Queen” on Minor Health Complaints in Healthy Adults: A Randomized, Double-Blind, Placebo-Controlled Study"

_nutrients, 2022, doi:10.3390/nu14122446_

Round 1

Reviewer 1 Report

The authors present a manuscript describing the study of the Shadow Queen potato variety effects in psychological welfare and other biological markers. The English is good, with only minor mistakes that could be quickly fixed in another proof read. The methodology employed is scientifically sound. The article was interesting and showed a possible uncommon effect for a specific potato variety consumption. Despite the small number in test subjects, the changes in the anxiety and stress responses could really be related to the nutrients of the studied object, but with the present proposed study that cannot be affirmed with certainty. That being said, I was skeptical of the claim made in the abstract, however, the authors made it clear that these results only hint at an “anti-depressive and anti-stress” effect and are not definitive. I only further suggest (along with the specific suggestions below) that the abstract conclusion should be rewritten in order to reflect this uncertainty, as the manuscript’s conclusion does.

“[…] test potato cv. SQ containing 45 mg anthocyanin. […]” The phrasing is unclear if this anthocyanin content was measured or added to the potato, maybe a rephrasing would improve clarity

“pain in the flesh” This term does not make sense

“average annual losses were $1165, $520, and $3055” losses per employee?

Table 1. How did the authors retrieve the nutritional information? The anthocyanin content was determined by which method? This information should be present in text

The methodology section should clarify if the t-student test was performed to verify the difference in averages between the test and control groups

Author Response

Thank you very much for your careful peer review. We have made the following corrections as indicated below, and we appreciate your re-review.

The authors present a manuscript describing the study of the Shadow Queen potato variety effects in psychological welfare and other biological markers. The English is good, with only minor mistakes that could be quickly fixed in another proof read. The methodology employed is scientifically sound. The article was interesting and showed a possible uncommon effect for a specific potato variety consumption. Despite the small number in test subjects, the changes in the anxiety and stress responses could really be related to the nutrients of the studied object, but with the present proposed study that cannot be affirmed with certainty. That being said, I was skeptical of the claim made in the abstract, however, the authors made it clear that these results only hint at an “anti-depressive and anti-stress” effect and are not definitive. I only further suggest (along with the specific suggestions below) that the abstract conclusion should be rewritten in order to reflect this uncertainty, as the manuscript’s conclusion does.

  1. “[…] test potato cv. SQ containing 45 mg anthocyanin. […]” The phrasing is unclear if this anthocyanin content was measured or added to the potato, maybe a rephrasing would improve clarity

Answer: Thank you for your advice. The first phrase has been corrected to "naturally contains anthocyanins" so that it is clear that anthocyanins have not been added.

  1. “pain in the flesh” This term does not make sense 

Answer:  Thank you for your comment.  We corrected “pain in the flesh” to “joint pain”.

  1. “average annual losses were $1165, $520, and $3055” losses per employee?

Answer: Thank you for your comment. The text was not clear and has been revised as follows.

“A study of four Japanese pharmaceutical companies estimated the average annual economic losses due to medical/pharmaceutical expenses per employee, absenteeism, and presenteeism were $1,165, $520, and $3,055, respectively”

  1. Table 1. How did the authors retrieve the nutritional information? The anthocyanin content was determined by which method? This information should be present in text

Answer: In accordance with your suggestion, we have added the analytical methods for nutrients and anthocyanins as follows.

Anthocyanin content was measured by the colorimetric method, water by the atmospheric heating drying method, protein by the combustion method, fat by the acid hydrolysis method, ash by the direct ashing method, dietary fiber by the enzymatic-gravimetric method, and sodium by the atomic absorption spectrophotometry. The conversion factor of nitrogen-protein was set at 6.25. Carbohydrates were calculated as "100 - (protein + fat + ash)," NaCl as "sodium x 2.54," and energy using energy conversion factors (protein: 4, fat: 9, carbohydrates: 4, and dietary fiber: 2).

  1. The methodology section should clarify if the t-student test was performed to verify the difference in averages between the test and control groups

Answer: The statistical process is summarized in section 2.10. Statistical Analysis, in which the Student-t test is described.

Reviewer 2 Report

The paper is very good. I don't have any additional comments. It can be accepted in present form.

Author Response

Thank you very much for your careful peer review.